# Comparing the Informative Value of 2-Minute Segments of the 6-Minute Walk Test: Insights into a Prospective Study on Parkinson’s Disease

**DOI:** 10.3390/s25227076

**Published:** 2025-11-20

**Authors:** Rosalia Zangari, Laura Brighina, Andrea Pilotto, Greta Carioli, Vincenzo D’Agostino, Armando Oppo, Andrea Rizzardi, Alessandro Padovani, Francesco Biroli, Dario Alimonti

**Affiliations:** 1FROM—Research Foundation Bergamo Hospital—ETS, 24127 Bergamo, Italy; gcarioli@fondazionefrom.it (G.C.); fbiroli@fondazionefrom.it (F.B.); 2Department of Neurology, Fondazione IRCCS San Gerardo dei Tintori, 20900 Monza, Italy; laura.brighina@irccs-sangerardo.it (L.B.); v.dagostino2@campus.unimib.it (V.D.); armando.oppo@gmail.com (A.O.); 3Neurology Unit, Department of Clinical and Experimental Sciences, University of Brescia, 25123 Brescia, Italy; andrea.pilotto@unibs.it (A.P.); a.rizzardi005@unibs.it (A.R.); alessandro.padovani@unibs.it (A.P.); 4Neurology Unit, Department of Continuity of Care and Frailty, ASST Spedali Civili of Brescia, 25123 Brescia, Italy; 5Laboratory of Digital Neurology and Biosensors, University of Brescia, 25123 Brescia, Italy; 6Center for Alzheimer Research, Division of Clinical Geriatrics, Department of Neurobiology, Care Sciences and Society (NVS), Karolinska Institutet, SE-171 77 Stockholm, Sweden; 7Brain Health Center, University of Brescia, 25123 Brescia, Italy; 8Department of Neurology, Papa Giovanni XXIII Hospital, 24127 Bergamo, Italy; dalimonti@asst-pg23.it

**Keywords:** Parkinson’s disease, gait, 6-minute walk test, 2-minute walk test, wearable sensors, motor performance, disease progression

## Abstract

**Highlights:**

**What are the main findings?**
The first 2 min of the 6-min walk test (2’6MWT) correlate strongly with full 6MWT gait parameters, especially stride length and gait speed.Agreement between the 2’6MWT and full 6MWT remained consistent at 1- and 2-year follow-up, while the full test is needed to capture subtle fatigue-related changes.

**What is the implication of the main finding?**
The 2’6MWT can serve as a practical, time-efficient surrogate for gait assessment in patients with Parkinson’s disease.Using sensor-based gait analysis, clinicians and researchers can reliably monitor walking performance while reducing patient fatigue and clinical testing time, as both the 2-min and 6-min measures can be derived from a single walking session.

**Abstract:**

Gait assessment is key in Parkinson’s disease (PD), but the psychometric properties of common tests like the 6-Minute Walk Test (6MWT) are not fully established. Inertial Measurement Units (IMUs) offer objective gait measures, potentially reducing repeated testing. This study evaluated whether the resampling of the first 2 min of the 6MWT (2’6MWT) reflects full-test performance in 43 early-to-mild PD patients (median age 65) at baseline, 1-year, and 2-year follow-ups. A trunk-mounted IMU recorded distance covered, walking duration, stride length, gait speed, cadence, and symmetry. Analysis focused on participants with complete longitudinal data from a multicenter original cohort of 62. Stride length and gait speed (2’6MWT vs. 6MWT) demonstrated strong correlations (r > 0.98), near-perfect agreement, <5% error, and stability across follow-ups; cadence showed slightly more variability. The analysis of consecutive 2-min intervals of the 6MWT revealed stable stride length and gait speed, with modest decreases in distance and cadence over time. Exploratory associations of 2’6MWT and 6MWT with motor severity and cognitive status were consistent. These results indicate the 2’6MWT is a reliable, time-efficient alternative to the full 6MWT for assessing walking capacity in PD, especially in outpatient or fatigue-prone patients. The full 6MWT remains valuable for detecting subtle endurance- or fatigue-related changes.

## 1. Introduction

Gait impairment in people with Parkinson’s disease (PD) is an important indicator of both physical and cognitive health associated with decreased functional independence [1], reduced quality of life [2,3], and increased risk of falls [4]. Although gait assessment is part of the routine clinical assessment of PD patients in many clinical settings, the reliability and validity of the different walk assessments have not been thoroughly investigated in this population. Moreover, these gait assessment instruments vary widely in their levels of administration complexity.

Walking tests, such as the 2-Minute Walk Test (2MWT) [5], the 6-Minute Walk Test (6MWT) [5] and the 10-Meter Walk Test (10MeWT) [6], are similar assessment tools originally developed in clinical contexts to evaluate different aspects of gait function—including functional mobility, endurance, and gait speed—in patients with impaired physical performance during rehabilitation [7] and in patients with cardiovascular or pulmonary diseases [5,8,9]. In PD, these tests have been adapted and are commonly used to evaluate gait disturbances, but also functional capacity as potential proxies of disease progression.

Among them, the 6MWT stands as a widely utilized and validated tool for assessing walking capacity in older adults with PD [10]. During the test, participants walk back and forth along a 30-m corridor for six minutes at a comfortable, self-selected pace [9].

The 2MWT and 10MeWT have been proposed as shorter alternatives to reduce the burden of assessment time or fatigue: the 2MWT measures the distance covered in two minutes, while the 10MeWT evaluates gait speed over a 10-m course, typically performed at both comfortable and maximal speeds.

In people with PD, the applicability of the 6MWT can be limited by time constraints for both examiners and patients, leading several protocols to adopt the 2MWT as a shorter alternative [11].

However, gait patterns in PD present unique challenges. Gait speed over short distances may differ between people with mild and moderate PD severity, and tends to decline over the course of the 6MWT, with participants maintaining only ~76% of maximal gait speed [12], largely due to hypokinesia, raising concerns about information redundancy [13]. It remains unclear whether—and how—gait performance measured across the full 6MWT changes with disease progression.

Furthermore, although shorter walk tests such as the 2MWT and 10MeWT are increasingly used in PD, their psychometric properties have not yet been fully established for this population [14].

Evidence from other neurological populations, including people with stroke and spinal cord injury, suggests that the 10MeWT tends to overestimate gait speed measured with the 6MWT [15,16,17]. In studies of populations with other neurodegenerative and neuromuscular diseases (other than PD), the 2MWT has been shown to provide comparable information to the 6MWT [18,19,20]. Specifically, the distance covered in the first 2 min of the 6MWT has been shown to closely match that of the standalone 2MWT, with gait speed stabilizing thereafter and distances remaining consistent over the final 4 min [18,21]

Recent advances in wearable sensor technology now enable detailed gait analysis during walking tests. The 6MWT performed with wearable inertial devices containing accelerometer and gyroscope sensors, offers a clinically feasible method for real-time, objective assessment of gait quality in patients with PD [13,22]. The use of these devices enables the assessment of spatiotemporal gait characteristics (e.g., gait speed, stride length), enhancing the evaluation of motor performance and confirming the test’s reliability and validity even over shorter durations [12,13,23], without requiring separate retesting.

Given the similar setup of the 2MWT and 6MWT, it may be possible to extract 2-min gait data from a standard 6MWT, thereby reducing extra testing. To our knowledge, no prior study has systematically compared sensor-derived gait parameters from the first 2 min of the 6MWT (the “*2’6MWT*”) versus the full 6 min test in people with PD.

By deriving both the 2 min and 6 min measures from a single walking session, this method potentially reduces participant burden and fatigue while allowing direct within-session comparison of the same walking performance at different time points. This methodological approach differs fundamentally from conducting two separate tests (a standalone 2MWT followed by a separate 6MWT).

This study aims to assess the informativeness of a core set of gait parameters (specifically, walking duration (min), distance covered (m), stride length (m), mean gait speed (m/s), cadence (steps/min), and general symmetry index) captured by a wearable sensor during the first 2 min of the 6-Minute Walk Test (2’6MWT), compared to those obtained over the full 6 min duration (6MWT), in patients with PD. The study employs a longitudinal design (assessments at baseline, 1-year, and 2-year follow-up) to examine within-subject changes in gait performance over time, thereby extending previous research.

As the first study to use this single-session approach in PD, we interpret findings with caution and discuss implications for future validation.

## 2. Materials and Methods

### 2.1. Study Design and Patient Characteristics

Data from the first 2 min of the 6-Minute Walk Test (6MWT) were analyzed as part of an exploratory analysis within a multicenter, observational study on PD, Gait Characteristics and Cognitive Evolution in Parkinson Disease (GECO-PARKINSON STUDY, NCT04297800).

The analysis presented here was restricted to data from two (BG, MB) of the three recruitment centers [the BG Center (Bergamo Hospital), the MB Center (Monza Hospital) and the BS Center (Brescia Hospital)], for which complete data across all time points (baseline, 1- and 2-year follow-up) were available. The final sample comprised 43 participants (Table 1). A sensitivity analysis was conducted to estimate the minimum detectable within-subject effect size given the available sample (*n* = 43, α = 0.05, power = 0.80). The study was adequately powered to detect within-subject effects of at least moderate magnitude (Cohen’s d ≥ 0.44), based on a paired-samples *t*-test.

Of these, 40 participants (93%) completed the 1-year and 35 participants (81%) completed the 2-year follow-up assessments.

No significant differences were found between this subset and the overall original cohort (n = 62) in terms of demographic or clinical characteristics (Appendix A).

The participants were included if diagnosed with idiopathic PD (MDS criteria) [24]. The inclusion criteria for the study were: (a) age 55–74 years, of either sex, (b) independent walking without assistive devices and no major medical conditions affecting walking ability, (c) Hoehn and Yahr stage (H&Y) 2–3 (stage 1 excluded: minimal unilateral symptoms and no noticeable gait disturbances), (d) stable response to anti-Parkinson’s medications for the 3 months prior to the recruitment moment and e) Mini-Mental State Examination (MMSE) > 24/30 total score to exclude cognitive deficits potentially impacting on mobility.

Exclusion criteria included other neurodegenerative parkinsonian syndromes, secondary parkinsonism, and conditions affecting cognitive or motor function, namely major psychiatric disorders, autonomic dysfunction, orthopedic issues, organ failure (Cumulative Illness Rating Scale > 2), or substance abuse.

The study protocol was approved by the Ethics Committees of all participating centers (see Institutional Review Board Statement), and all participants provided written informed consent.

General demographic information (age and gender), characteristics of the disease (i.e.,: years since diagnosis, initial disease severity according to H&Y) and health information were collected at baseline (Table 1). The investigators from the centers were trained all together in conducting the assessments to optimize the standardized collection of all outcome measures.

### 2.2. Clinical and Sensor-Based Assessments

Assessments were conducted during annual visits for three consecutive years (i.e., baseline, 1-year, and 2-year follow-up). At each visit, participants underwent a clinical evaluation and a gait assessment using a wearable inertial measurement unit (IMU) sensor device (Figure 1).

The sensor was attached to the participant’s waist with a semi-elastic belt and positioned over the sacral area (S1–S2). The tests were conducted by the same examiner to avoid inter-rater bias. All participants received dopaminergic treatment (including L-Dopa and dopamine agonists) and were evaluated during their ON phase.

The 6MWT [9] was used to determine walking ability and to measure spatiotemporal variables of walking over a longer distance. In the 6MWT, participants walked at a self-selected pace along a 10-m indoor corridor. Traffic cones were placed at both ends of the path to mark the turning points. The testing location remained the same for all tests.

The software automatically computes a predefined core set of spatiotemporal gait parameters, providing a report with summary metrics for each 6MWT trial, including walking duration, distance, stride length, mean gait speed, cadence, a general symmetry index and number of gait stops. The present analysis focused only on this core set of parameters. The software also provides raw data with detailed summaries for each segment of the walk (called “patch,” a time window of collected data), including both spatiotemporal and step-by-step parameters. Gait parameters were extracted from the first 2 min of the 6MWT, referred to as the “2’6MWT” to distinguish it from previous studies where the 2MWT and 6MWT were performed separately. Raw 6MWT data were resampled to match the software report values, and after verifying this correspondence, the 2’6MWT parameters were calculated for each patient. In addition to the initial 2-min segment (2’6MWT, Segment 1), the subsequent 2-min segments of the 6MWT (Segments 2 and 3) were also extracted for analysis.

Figure 1 provides a schematic overview of the clinical and sensor-based assessments conducted in this study. The image was generated with the assistance of generative AI and subsequently edited to include schematic labels relevant to this paper, as well as a depiction of the sensor positioned on the sacral area.

The severity of the disease (H&Y) was reassessed at each visit. The AR (Akinetic-Rigid) and TD (Tremor-Dominant) subtypes were determined based on the patients’ clinical records, reflecting a comprehensive clinical evaluation. Patients were classified as TD if tremor was the predominant motor feature, and as AR if bradykinesia and rigidity were predominant. The motor subscale of the Movement Disorder Society Unified Parkinson’s Disease Rating Scale (MDS-UPDRS) Part III (Motor Exam) was used to assess motor symptoms. Postural Instability and Gait Difficulty (PIGD) and (Freezing of gait (FOG) were quantified using UPDRS Part III (items 3.9, 3.10, 3.11, 3.12, 3.13) and item 3.11, respectively. FOG was evaluated during the clinical assessment through the UPDRS Part III “Freezing of Gait” item. Fall history was evaluated using a self-reported question related to the month preceding the motor assessment. The International Physical Activity Questionnaire (IPAQ) was used to assess the level of physical activity. Cognitive function was assessed with the MMSE and the Montreal Cognitive Assessment (MoCA), adjusted for education level and age.

L-Dopa Equivalent Dose (LED) and Levodopa Equivalent Daily Dose (LEDD) were recorded. LED refers to the equivalent dose of a single medication based on standardized conversion factors, LEDD represents the total daily dose of all antiparkinsonian medications [25].

### 2.3. IMU-Based Analysis of Spatiotemporal Gait Parameters

Each patient was equipped with an IMU, the BTS© G-Walk^®^, a portable, wearable sensor with wireless data transmission (G-Sensor 2; BTS Bioengineering S.p.A., Garbagnate Milanese (MI), Italy).

The inertial platform is equipped with 4 Sensor Fusion technology that consists of a triaxial accelerometer (16 bit/axis) with multiple sensitivity levels (±2, ±4, ±8, ±16 g), a triaxial gyroscope (16 bit/axis) with multiple sensitivity levels (±250, ±500, ±1000, ±2000°/s), a triaxial magnetometer (13 bit, ±1200 µT), and a global positioning system receiver. The device was placed on the waist of the participant at the sacral level with a semi-elastic belt and recorded the acceleration data. All acceleration data were sampled at a frequency of 100 Hz, transmitted to a notebook via Bluetooth and processed with a specific software (G-Studio, version 3.3.22.0). If the Bluetooth sensor was out of range and recorded a value of 0, the corresponding data points were excluded from the analysis.

Data were recorded and processed using the manufacturer’s G-Studio software according to the standard protocol. The software automatically segments the recorded walk into stable gait paths, excluding the first and last paths, which correspond to the initial acceleration and final deceleration phases, respectively, as part of its default processing. Steps associated with discontinuities or irregularities during walking are also removed. Consequently, only steady-state gait segments are included in the computation of spatiotemporal parameters, preventing bias from acceleration or deceleration phases (according to the device specifications, e.g., as reported in *G-Walk User Manual*, v.7.0.0). This default processing is already reflected in both the raw data and the clinical report generated by the system.

The exact algorithms of the G-Walk are proprietary and not publicly available. However, according to the manufacturer’s technical specifications, stride length (m) is defined as the distance between two consecutive heel-strikes of the same foot, calculated as the product of stride-specific velocity and stride duration (Figure 1). Gait speed (m/s) is obtained by integrating the accelerometer signal over each walking trial.

The G-Walk system has been previously validated against the gold standard in both healthy adults and patients with PD, showing good agreement for spatiotemporal parameters such as stride length, gait speed, cadence and stride duration [26,27].

### 2.4. Statistical Analysis

Patients’ characteristics were summarized using descriptive statistics; continuous variables were reported as median and interquartile range (IQR) according to the non-normal distribution of the data (Shapiro–Wilk test) and categorical ones as frequency and percentage. Differences between independent groups were evaluated using the Mann–Whitney U test for continuous variables and the Chi-square or Fisher’s exact test for categorical ones. For paired comparisons between 2’6MWT and 6MWT parameters, the Wilcoxon signed-rank test or the Sign test was applied, as appropriate. For comparisons across the three consecutive 2-min segments (segments 1–3) of the 6MWT, the Friedman test was used to assess overall effects, with Kendall’s W to quantify effect size. When significant, pairwise comparisons were explored using Bonferroni-adjusted Wilcoxon signed-rank post hoc tests.

Spearman’s correlation was used to evaluate (1) agreement between 2’6MWT and 6MWT gait parameters and (2) their associations with baseline clinical variables. Unadjusted univariate linear regressions quantified how 2’6MWT predicted corresponding 6MWT and clinical measures. Variables showing at least 2 significant correlations were further analyzed in models adjusted for recruitment center, age or sex.

Concordance between the gait parameters obtained during 2’6MWT and 6MWT was assessed using the Bland–Altman method. Inter-rater reliability was evaluated by calculating the Concordance Correlation Coefficient (CCC). To assess between-method differences (bias), percentage errors (PE) were also calculated.

Outliers were identified using the Tukey method. Sensitivity analyses excluding these values, as well as robustness checks using Passing–Bablok regression, were performed to confirm the stability of results, as a robust non-parametric method.

Only complete and continuous recordings throughout the 6MWT were included in the analyses to ensure accurate gait parameter estimation.

Statistical analyses were performed using R (version 4.3.3) and Stata software (version 16; StataCorp LP, College Station, TX, USA).

## 3. Results

### 3.1. Participants’ Characteristics

Table 1 summarizes the demographic and clinical data of the 43 patients with idiopathic PD at baseline evaluation.

The median age of the participants was 65.0 years (IQR: 60.0–69.5, range: 55.0–74.0) with 28 (65.1%) male and 15 female participants, resulting in a male-to-female ratio of 1.87:1. Most patients had a disease severity corresponding to stage 2.0 on the H&Y scale, with 81.4% at stage 2 and 18.6% at stage > 2. Most patients presented with the akinetic-rigid subtype (58.1%). Median disease duration was around 4 years and median age at diagnosis was around 61 years. The median UPDRS Part III score, which assesses motor impairment, was 22, indicating moderate motor dysfunction. Gait and balance were generally preserved during the trial, with a median PIGD score of 2.

No participant exhibited FOG episodes during the clinical assessment (score of 0 on the UPDRS Part III), and no falls were reported by participants in the month preceding the motor assessment. Included patients did not present any comorbidities, such as orthopedic issues, major psychiatric disorders, or organ failure, ensuring that gait alterations were primarily related to PD.

Cognitive function was preserved at baseline: all participants scored above the established cut-off on the adjusted MMSE (≥24), and 65.1% also exceeded the MoCA cut-off (≥26). Notably, 65% of patients had an educational level of less than 12 years. Regarding treatment, the median L-Dopa equivalent dose (LED) was 300.0 mg (IQR: 250.0–462.5), while the overall LEDD was 450.0 mg (IQR: 320.0–730.0).

The comparison between the centers on demographic and clinical factors is described in Appendix A).

### 3.2. Comparison of 2’6MWT and 6MWT Parameters

Data showed distinct patterns between gait parameters measured during the first 2 min and the full 6 min of the 6MWT (Table 2). Overall, the test was performed in 6.04 min (IQR: 6.02–6.15), while for the 2’6MWT 2.03 min (IQR: 1.99–2.04) were considered. Each participant was able to finish the 6MWT task, except for one patient who only covered 80.6 m (2.39 min) and was excluded from the subsequent analysis, resulting in a sample of 42 patients for the outcome analysis. No gait stops (i.e., interruptions of walking) were detected by the sensor-based 6MWT recording system.

The observed ranges for gait parameters (Table 2) during the 6MWT were as follows: step length 0.8–2.5 m (median 1.3 m), gait speed 0.7–1.8 m/s (median 1.2 m/s) and cadence 61.1–134.4 steps/min (median 115.1 steps/min). Fewer than 5% of values were identified as outliers; their exclusion did not substantially affect the median values or the results of the statistical tests, and therefore all values were retained in the analysis.

The distance covered by the participants during the 2’6MWT was correlated with the distance covered during the entire 6MWT (Spearman, r = 0.882, *p* < 0.001, Table 2). No significant intra-patient difference in stride length was found between the 2’6MWT and 6MWT. A strong correlation (r = 0.988, *p* < 0.001) and a linear regression with a slope close to 1 and an intercept close to 0 confirmed the agreement between the parameters (Table 2, Figure 2).

Despite a strong correlation (r = 0.985, *p* < 0.001) and consistent regression results, gait speed showed a significant intra-patient difference (*p* = 0.015). Similarly, cadence showed significant differences between the 2- and 6 min tests, and although a strong correlation was observed, the regression analysis indicated a lack of agreement between the parameters (Table 2). Similar results to those obtained with standard linear regression were observed using the Passing–Bablok regression model (Appendix A).

In addition, the exploratory analysis of consecutive 2-min segments of the 6MWT (Appendix A) revealed a modest but significant reduction in distance covered and cadence over time, whereas test duration, stride length and gait speed remained stable. Specifically, patients walked a slightly greater distance and maintained a higher cadence during the initial 2 min compared with later phases (distance: χ^2^ = 11.9, *p* = 0.003; cadence: χ^2^ = 23.7, *p* < 0.001).

The CCC values were equal to over 0.98 (95% CI: 0.96–0.99), mean differences around 0 and limits of agreement were within acceptable ranges for stride length and gait speed (Table 3 and Figure 2), while cadence also showed strong agreement but a mean difference of −1.18 and a slightly wider range of variation. The PE between the 2’6MWT and 6MWT was 1.98% for stride length, 2.64% for gait speed, and 1.5% for cadence.

Table 4 presents an exploratory analysis of the bivariate correlations between clinical variables and gait parameters from both 2’6MWT and 6MWT.

Although exploratory in nature (rho range: ±0.30–0.49, indicating moderate correlations, *p* < 0.05), the interpretation focused on variables showing at least two corresponding significant correlations across both tests. Within this criterion, greater motor impairment (higher PIGD scores), lower physical activity levels (lower IPAQ scores), and reduced cognitive performance (lower adjusted MMSE, while MoCA did not show significant associations in this cohort) were associated with reduced walking performance across 2’6MWT and 6MWT. After adjustment for sex and age (for MoCA and PIGD) or sex alone (for MMSE), these associations remained significant for MMSE and PIGD, whereas MoCA continued to show no robust correlations (secondary and confirmatory analyses). Furthermore, regression models including center as a covariate indicated that the relationships between MMSE or PIGD and gait were largely unaffected (secondary and confirmatory analyses).

### 3.3. Temporal Stability and Concordance of Gait Metrics in Repeated Assessments

The comparative analysis between the first 2 min and the entire 6MWT showed consistent results for the gait parameters over time (1-year, 2-year after baseline assessment), see supplemental results (Appendix A). Significant differences were observed in cadence and gait speed (the latter only for the 1-year assessment), with lower values recorded for the 6MWT (*p* < 0.05). However, stride length remained stable across time points. Correlations (r > 0.90) were observed for all parameters, particularly for stride length (r = 0.93–0.98) and gait speed (r = 0.95–0.97). CCC indicated almost perfect agreement between measures (CCC > 0.93). Bland–Altman analyses showed minimal average differences for stride length and gait speed.

The analyses focused on the relationship between 2’6MWT and 6MWT, regardless of longitudinal clinical changes. However, detailed information on clinical variability over time is provided in the Appendix A.

## 4. Discussion

This study represents the first systematic analysis of sensor-derived gait parameters in patients with idiopathic PD, comparing the first 2 min of the 6-Minute Walk Test (2’6MWT) with the full 6 min duration (6MWT). It introduces a novel and practical methodological approach by deriving both measures from a single walking session, aiming not only to reduce participant burden and fatigue but also to enable a direct within-session comparison of early versus overall walking performance.

At baseline, the cohort consisted of patients at early-to-mild stage PD (H&Y ≤ 2.5; 1 patient at stage 3, stage 1 excluded), with a predominance of the akinetic-rigid subtype (58.1%) [28], moderate motor dysfunction, and preserved cognitive function [29], thereby ensuring homogeneous testing conditions. All patients were assessed in the ON state under stable dopaminergic treatment, ensuring consistency in motor evaluation.

The main findings of this study indicated that the 2’6MWT served as a reliable surrogate for the full 6MWT in the assessment of gait parameters, showing strong correlation and agreement for stride length and gait speed, and demonstrating good reproducibility over time. Although stride length and gait speed were practically interchangeable between the 2’6MWT and 6MWT, cadence showed small systematic differences. Overall, relative differences between the 2’6MWT and 6MWT remained within clinically acceptable limits (≈5%). Distance covered was proportionally related, reflecting the shorter test duration.

The use of a wearable sensor can streamline 6MWT assessments by providing accurate, consistent data from a single session, supporting the notion that a 2’6MWT derived from the full 6MWT may be sufficient. Sensor-based gait parameters have demonstrated good test–retest reliability in healthy individuals, in patients with multiple sclerosis and post-stroke [30,31,32], and in PD [13], confirming the robustness of this methodological approach and reducing the need for repeated assessments.

Across all time points evaluated in this study (baseline, 1-year, and 2-year follow-up), gait parameters recorded during the first 2 min were consistent with those observed over the full 6MWT. While cadence and gait speed showed slightly lower values, stride length remained stable across time points, reflecting preserved walking biomechanics. Strong correlations and near-perfect agreement across parameters further support the reliability of the 2’6MWT (stride length and gait speed) as a valid proxy for the full 6MWT.

Given the limited evidence available in PD, findings from other populations with neurological or gait impairments (summarized below) may help contextualize the present results on the 2’6MWT, although gait test properties should ultimately be confirmed within the target population.

In this cohort, the distance covered during the 2’6MWT and during the whole 6MWT were highly correlated, as well as in other neurological or motor-impaired populations (such as chronic multiple sclerosis [33], frail older adults with dementia [34], stroke [21], lower limb amputations [35]).

Stride length was comparable between 2’6MWT and 6MWT, in line with previous research demonstrating that stride length remains stable even when 6MWT data are segmented into three distinct two-minute intervals, except for the total number of strides completed [23,36].

Gait speed was greater during the initial 2 min of the 6MWT, without influencing its correlation with the total 6MWT performance. Participants typically start at a faster pace and gradually slow down as fatigue develops toward the end of the test [37]. This decrease in gait speed during prolonged walking may also be influenced by the sequence effect, in which a progressive reduction in step amplitude and speed during repetitive movements reflects motor control deficits rather than muscular fatigue [38,39].

Similarly, cadence showed increased variability during the first 2 min of the 6MWT. This effect likely reflects an acclimation period as participants adjusted to the test, with variability diminishing in subsequent segments, suggesting a stabilization of gait patterns [36]. Notably, cadence values recorded in the 2’6MWT remained strongly correlated with those of the entire 6MWT.

Analyzing the 6MWT in three consecutive 2 -min segments revealed a slight decrease in distance and cadence during the first segment, while stride length and gait speed remained stable. This pattern suggested an initial adjustment period followed by steady performance and indicates a mild fatigability effect. The reduction was mainly driven by cadence, consistent with prior studies identifying step frequency as a sensitive marker of endurance-related gait deterioration in patients with chronic conditions or neurological impairments [40].

Speculatively, participants appeared to walk faster during the initial 2 min of the 6MWT, but this difference was not statistically significant when the test was analyzed in consecutive 2-min segments. This likely reflects a combination of statistical factors—small differences are detectable in aggregate comparisons but not after segmental post hoc adjustments—and clinical characteristics, such as an initial pacing effect followed by stabilization of gait speed to sustain effort throughout the test.

These results also align with the broader body of evidence suggesting that shorter walk tests can provide reliable estimates of walking capacity. In this context, the 2MWT has gained increasing attention as a time-efficient alternative to the 6MWT [40]. Although originally less studied in PD, the 2MWT has been validated in neurological populations, including stroke [21,41], multiple sclerosis [18,42], and spinal cord injury [43]. Notably, the 2MWT appears particularly advantageous in these cohorts with greater fatigability or disability, for whom completing six minutes of walking may be challenging.

In these studies, 2MWT and 6MWT performances were highly correlated, consistent with the present cohort (r = 0.88–0.99, *p* < 0.001), reinforcing the view that a shorter and more convenient test can serve as a reliable substitute for the 6MWT, especially in time-constrained clinical settings. Notably, in the cohort, all participants successfully completed the first 2 min of the test (at baseline). Although one participant was not included in the analyses because they discontinued after this initial phase, this observation further supports the feasibility and clinical relevance of shorter walking assessments even among individuals with greater motor limitations [38]

Although exploratory to provide context for the gait parameters, the associations with key clinical variables—including PIGD scores, IPAQ scores, and cognitive status—were consistent across both the 2’6MWT and 6MWT.

In summary, while assessing performance over longer durations remains important in conditions such as PD to capture overall trends and disease progression, the first 2 min of the 6MWT (2’6MWT) provide an efficient and practical alternative, with stride length and gait speed largely preserved. This approach is particularly valuable in outpatient settings, where time is often a limiting factor, and in patients susceptible to fatigue or stress, offering a valid substitute for the full 6MWT without significant loss of information. Clinically, these findings have a dual implication: the 2’6MWT is sufficient for monitoring global walking capacity, as reflected by gait speed and stride length, and can serve as a reliable surrogate in resource-limited conditions, whereas the complete 6MWT remains important for detecting subtle endurance- or fatigue-related changes.

Overall, these results support the use of the 2’6MWT as a representative segment of the full 6MWT test, potentially reducing the need for a separate 2MWT. However, this interpretation is preliminary and does not imply that the 2MWT can be entirely replaced by the 2’6MWT. Rather, the 2’6MWT appears to provide a reliable reflection of the full 6MWT. This approach may be particularly valuable in time-constrained clinical settings or for patients with limited endurance, offering meaningful gait information while minimizing patient burden.

This study had some limitations. First, due to COVID-19 pandemic countermeasures, some assessments were postponed, limiting the number of evaluations.

Second, the first 2 min of the 6MWT were analyzed instead of conducting a separate 2MWT, minimizing patient burden and optimizing adherence to the protocol.

Although this approach enhances feasibility in clinical settings, it is still important to acknowledge that, in theory, performing the tests separately could introduce sequence effects, as well as differences in motivation or pacing.

Third, this study focused on a core set of temporal gait parameters—walking duration, distance covered, stride length, gait speed, cadence, and general symmetry index—that were reliably captured by the G-Walk system over the entire 6MWT. Some limitations in parameter assessment should be noted: stride-level metrics such as gait variability and arrhythmicity were not available; step-by-step parameters were not analyzed, as there was no correspondence with the summary report; the general symmetry index was calculated only for the complete walking trial, as analyzing 2 -min intervals would have required reprocessing the raw data, potentially affecting accuracy.

Technical considerations of the IMU system: Trunk-mounted IMUs may show systematic differences in certain gait phase–related parameters (e.g., stance, swing, single and double support) compared with gold-standard optical motion capture systems [26,27]. While IMUs reliably capture spatiotemporal gait parameters such as stride length and gait speed, they may be less accurate for detailed phase-specific analyses. Therefore, caution is recommended when using IMUs for stride-by-stride or phase-specific gait assessments. Moreover, while a single IMU provides reliable core gait parameters, multi-sensor systems could capture additional features such as turning dynamics, asymmetry, and stride-level variability, offering a richer characterization of gait.

Sample characteristics and generalizability: The cohort comprised early-to-mild stage PD patients in the “ON” state, none of whom exhibited gait interruptions or freezing during the 6MWT. Fatigue and freezing of gait are significant determinants of performance in longer walking tests in more advanced stages of the disease. Therefore, further validation of the 2’6MWT in patients with more advanced PD, higher fatigue burden, or frequent freezing episodes is warranted, although this was beyond the scope of the present study.

Center-related and procedural considerations: Demographic and clinical comparisons between centers showed no significant differences; however, variations in physical activity levels and daily L-Dopa intake were observed. Although the technical methodology (including the positioning of the G-Walk sensor and the standardized execution of the 6MWT protocol) was the same at both centers, slight differences in calibration or distance covered settings may have influenced the cadence measurements, which then translated into differences in gait speed and distance covered.

## 5. Conclusions

This study demonstrated that the 2’6MWT can reliably assess gait parameters, such as stride length and gait speed, in patients with PD. It offered a practical, time-efficient alternative to the full 6MWT for both assessment and monitoring of gait. The findings support the use of the first 2 min of the 6MWT as a valid surrogate for the entire test, particularly in clinical settings with time constraints or in patients susceptible to fatigue. This approach shows promise in improving gait assessment in PD while reducing participant fatigue and enhancing overall compliance.

## Figures and Tables

**Figure 1 sensors-25-07076-f001:**
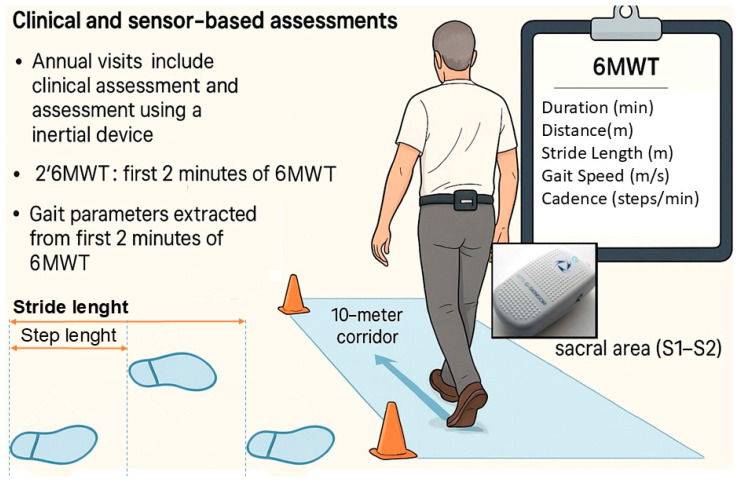
Positioning of G-Walk sensor and gait protocol for 6MWT. The figure provides a schematic over-view of the procedures. For the 6MWT, the box in the upper right corner reports the monitored parameters.

**Figure 2 sensors-25-07076-f002:**
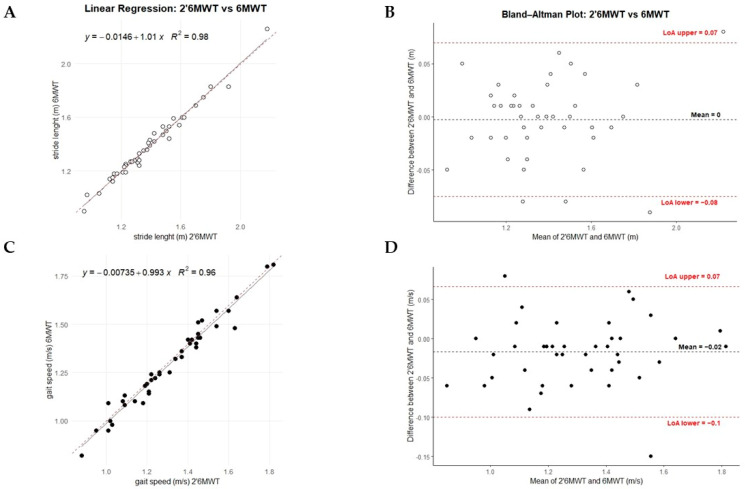
Performance of 2’6MWT vs. Total 6MWT. Panels (**A**,**C**) display each participant’s 2’6MWT (x-axis) and 6MWT (y-axis) gait parameters (stride length and gait speed), along with a dotted reference line (y = x) for comparison; the regression equation estimates the relationship between the 2’6MWT and 6MWT. Panels (**B**,**D**) show Bland–Altman plots between 2 and 6 min, depicting the mean and differences for each participant’s stride length and gait speed data. Plot colors: stride length (white) and gait speed (black).

**Table 1 sensors-25-07076-t001:** Participant Demographics and Baseline Characteristics.

Variables	Value
*N*	43
** *Demographic and Clinical Characteristics* **	
Age (y), *median (IQR)*	65.0 (60.0–69.5)
Sex, (male, *n* (%))/female	28 (65.1)/15
Height (m), *median (IQR)*	170.0 (163.5–178.0)
Weight (kg), *median (IQR)*	72.0 (62.5–82.0)
BMI (kg/m^2^), *median (IQR)*	24.8 (23.1–27.1)
Hoehn and Yahr stage, *median (IQR)*	2.0 (2.0–2.0)
HY stage 2, *n* (%)	35 (81.4)
HY stage 2.5, *n* (%)	7 (16.3)
HY stage 3, *n* (%)	1 (2.3)
Disease duration (y), *median (IQR)*	4.3 (2.1–8.0)
Age at diagnosis (y), *median (IQR)*	61.2 (56.0–64.0)
Age of onset (y), *median (IQR)*	59.4 (55.3–63.2)
Subtype (AR/TD), *n* (%)	25 (58.1)/18 (41.9)
LEDD (mg), *median (IQR)*	450.0 (320.0–730.0)
LED (mg), *median (IQR)*	300.0 (250.0–462.5)
** *Motor and Cognitive Assessments* **	
UPDRS III, motor exam, *median (IQR)*	22.0 (18.0–28.5)
PIGD ^1^, *median (IQR)*	2.0 (1.0–3.0)
FOG ^1^, *median (IQR)*	0
*IPAQ (Low/Mod/High)* ^2^, *n* (%)	9 (20.9)/ 23 (53.5)/11(25.6)
Adjusted MMSE (range, 0–30), *median (IQR)* ^3,4^	28.0 (27.0–29.1)
Adjusted MMSE ≥ 24, *n* (%)	43(100.0)
Adjusted MoCA (range, 0–30), *median (IQR)* ^3^	27.0 (23.5–28.0)
Adjusted MoCA ≥ 26, *n* (%)	28 (65.1)

BMI, Body Mass Index. HY stage, Hoehn and Yahr stage. AR, Akinetic-Rigid; TD, Tremor-Dominant. UPDRS, Unified Parkinson’s Disease Rating Scale (Part III-Motor Exam, range, 0–132). UPDRS III subscores ^1^: Postural Instability and Gait Difficulty (PIGD) is quantified using (items 3.9, 3.10, 3.11, 3.12, 3.13), FOG (Freezing of Gait) corresponds to item 3.11. ^2^ IPAQ, International Physical Activity Questionnaire classification. MMSE, Mini–Mental State Examination. MoCA, Montreal Cognitive Assessment [Adjustment for ^3^ education level and ^4^ age]. Levodopa Equivalent Daily Dose—LEDD is the amount of the daily dopaminergic drugs. L-Dopa Equivalent Dose—LED is the amount of the daily L-Dopa intake. Continuous variables are expressed as median and interquartile range (IQR), categorical ones as count and percentage.

**Table 2 sensors-25-07076-t002:** Comparative Analysis of Gait Parameters at Baseline Evaluation: 2’6MWT vs. Total 6MWT.

Tests	2 min	6 min	*Paired Test*	*Spearman* *Correlation*	*Linear Regression*	
*6MWT*			*p*-value *	r (*p*-value)	Slope (95% CI)	Intercept
Distance covered (m)	120.4 (97.8–123.7)	335.4 (303.4–371.1)	-	**0.882 (<0.001)**	2.68 (2.30–3.05)	31.33
Stride length (m)	1.3 (1.2–1.5)	1.3 (1.2–1.5)	ns	**0.988 (<0.001)**	**1.0 (0.96–1.06)**	**−0.01**
Gait Speed (m/s)	1.3 (1.2–1.5)	1.2 (1.1–1.4)	**0.015**	**0.985 (<0.001)**	**0.99 (0.93–1.05)**	**−0.0**
Cadence (step/min)	117.8 (112.0–123.5)	115.90 (110.3–122.3)	**<0.001**	**0.985 (<0.001)**	0.94 (0.90–0.98)	5.66

The data of 2 and 6 min of 6MWT are given as median values with interquartile range (IQR). Significant results (*p*-value < 0.05) marked in bold. ns: non-significant. * Paired comparisons were performed using the Wilcoxon signed-rank or Sign test. Interpretation: |r| > 0.7: strong correlation; slope > 1: strong relationship; slope = 1 and intercept = 0: perfect linear relationship. Missing data: One patient unable to complete the 6 min of the test. Bold indicates parameters for which the 2-minute and 6-minute measurements showed a consistent relationship, being statistically significant in all three analyses

**Table 3 sensors-25-07076-t003:** Concordance and Reliability of Gait Parameters at Baseline Evaluation: 2’6MWT vs. Total 6MWT.

Gait Parameter	CCC (95% CI)	Bland–Altman
		Average Difference (95% CI of LoA)
Distance covered (m)	-	-
Stride length (m)	**0.99 (0.98–0.99)**	**−0.002 (−0.07–0.07)**
Gait Speed (m/s)	**0.98 (0.96–0.99)**	**−0.02 (−0.10–0.07)**
Cadence (step/min)	**0.99 (0.98–0.99)**	−1.18 (−4.69–2.33)

The data of 2 and 6 min of 6MWT are considered. CCC, Concordance Correlation Coefficient and corresponding 95% confidence interval (CI). LoA, Limits of Agreement. Significant results marked in bold. Interpretation: CCC: 0–0.90: poor agreement; 0.91–0.99: almost perfect agreement. Bland–Altman average difference: 0 perfect agreement; ≠0 lack of agreement. Missing data: 1 patient unable to complete the 6 min of the test. Bold indicates gait parameters showing consistent results between the two analyses.

**Table 4 sensors-25-07076-t004:** Correlation matrix.

Variables	2’6MWT				6MWT			
	Distance Covered	Stride Length	Gait Speed	Cadence	Distance Covered	Stride Length	Gait Speed	Cadence
Motor Subtype (AR/TD)		(±)0.31 *				(±)0.33 *		
Height		0.46 **			0.34 *	0.42 **		
BMI	(−)0.32 *							
Disease duration				0.49 **				0.49 **
UPDRS III (total score)	(−)0.31 *							
PIGD ^1^	(−)0.45 **	(−)0.33 *	(−)0.36 *		(−)0.44 **	(−)0.34 *		
IPAQ ^2^			(−)0.32 *	(−)0.32 *			(−)0.37 *	(−)0.32 *
Adjusted MMSE		0.38 *	0.37 *		0.38 *	0.37 *	0.35 *	
LEDD				0.47 **				0.46 **
LED		0.33 *				0.36 *		

The data of 2 and 6 min of 6MWT are considered. Unadjusted bivariate correlations. Correlation strength according to Cohen: Mild correlation, r = ±0.30–0.49, *p*-value < 0.05 *, <0.01 **. Only statistically significant *p*-values are reported. The magnitude of correlation is indicated. Motor Subtype (Akinetic-Rigid = 1, Tremor-Dominant = 2): Negative correlations indicate that the AR subtype is associated with better performance in gait parameters compared to the TD subtype. BMI, Body Mass Index. UPDRS, Unified Parkinson’s Disease Rating Scale. MMSE, Mini-Mental State Examination. Levodopa Equivalent Daily Dose—LEDD is the amount of the daily dopaminergic drugs. L-Dopa Equivalent Dose—LED is the amount of the daily L-Dopa intake. Variables such as age, gender, H&Y stage, and adjusted MoCA were also tested but did not yield significant results. Postural Instability and Gait Difficulty (PIGD) ^1^ is quantified using UPDRS III (items 3.9, 3.10, 3.11, 3.12, 3.13). ^2^ IPAQ, International Physical Activity Questionnaire classification.

## Data Availability

The original contributions presented in this study are included in the article. Further inquiries can be directed to the corresponding authors.

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
