# Peer review of "Comparing the Informative Value of 2-Minute Segments of the 6-Minute Walk Test: Insights into a Prospective Study on Parkinson’s Disease"

_sensors, 2025, doi:10.3390/s25227076_

Round 1
Reviewer 1 Report
Comments and Suggestions for Authors
Summary
This study investigates whether the first two minutes of the 6-Minute Walk Test (6MWT) are representative of the full 6MWT performance in individuals with early-to-moderate stages of Parkinson’s disease (PD). Using a G-WALK inertial sensor positioned on the lower back, gait parameters such as distance, stride length, cadence, and gait speed were analyzed in 43 PD patients at baseline and at one- and two-year follow-up intervals. The results demonstrate strong correlations between the 2MWT and the full 6MWT, particularly for stride length and gait speed, suggesting that the shortened test may serve as a practical and time-efficient alternative for clinical gait assessment in early-stage PD.
The paper is methodologically sound, clearly structured, and relevant to the journal’s scope. The topic fits well within Sensors’ focus on wearable technology and clinical applications. The work contributes to the growing literature on sensor-based gait analysis and the optimization of clinical testing protocols.
There are a few important points I suggest the authors to address before the paper can be published.
Introduction
The manuscript would benefit from a clearer articulation of how the present work advances beyond prior studies -beside the reduced duration of the test, for example, by emphasizing the longitudinal (1-2-year) aspect of the current work.
I suggest the authors to briefly explain why they extracted and analyze much less parameters than those used in previous studies who investigated the 6MWT in PD with identical set-up, e.g.,: Bailo, G., Saibene, F. L., Bandini, V., Arcuri, P., Salvatore, A., Meloni, M., ... & Carpinella, I. (2024). Characterization of walking in mild Parkinson’s disease: Reliability, validity and discriminant ability of the six-minute walk test instrumented with a single inertial sensor. Sensors, 24(2), 662.
Please briefly describe in the introduction the 6MWT/2MWT and 10 MWT -I mean briefly describe the procedure and the instructions to participants. Currently, it is not really clear for someone who is not a PD expert.
Methods
Number of participants is not reported -it is only mentioned in the abstract and a different number is reported in the Supplementary materials as ‘original cohort’. Please specify what this means and report it in the main text of the manuscript.
The sample size (n=43) is reasonable for exploratory work, yet a power analysis or sensitivity estimate would strengthen the justification.
Please move Table 1 at the beginning of the Methods, when participants are described and add IPAQ scores to it -or explain why you did not report them in the Table nor analyzed them. Add to this Table also the subjectively reported falls.
Line 211: unclear what are the “two methods” compared with the Bland-Altman method. From Figure 2 it seems this test was used to compare performance in the two tests’ length (6 vs 2 minutes long). Please clarify this in the methods.
I suggest to add to the methods the description of the correlations with baseline characteristics of the patients
Results
Why one participant was excluded based on distance covered? The authors say they excluded this patient because was not able to finish the task, but then only distance covered was reported. Please explain it further.
The authors use non-parametric tests and correlations appropriately, but the lack of correction for multiple comparisons (beyond Bonferroni for Friedman tests) may risk type I error inflation.
Why other collected baseline measures such as IPAQ scores were not correlated with the 2MWT and 6MWT? I suggest to include them as they may also impact the outcomes.
How AR and TD subtypes were defined? Please specify
No patient experienced freezing of gait during the gait trial? I suggest to specify it (and state whether your group did include freezers or not).
How many participants performed the test again at 1 year and at 2 years post baseline assessment? Please add this information, and also please report in the Table about individual baseline values, also available data for these later assessments -whatever you collected.
Please report confidence intervals for the mean difference resulting from the CCC analyses also in the main text and not just in Tables.
Discussion
Line 328: The authors state they tested patients at early stage of disease but they had patients also with H&Y score of 3 considered moderate stage of disease. Please correct.
I suggest to mention in the discussion that this shorter 2M test might be fine for early-stage PD or certain PD subtypes (e.g., non-freezers), but it might be insufficient to capture other aspects of the disease such as freezing of gait that becomes more common when the patient experience a little fatigue (i.e., with longer assessments). More advanced stages of PD may need another validation.
The study relies on a single IMU (G-WALK). Although this is consistent with prior literature, the reliability of single-sensor placement should be discussed in relation to multi-sensor systems that may capture turning dynamics or asymmetry and provide richer pool of gait parameters. I suggest to add this to the discussion because it would strengthen the translational implications for clinical deployment.
The findings show strong correlations between 2MWT and 6MWT parameters; however, correlation does not necessarily imply interchangeability. I suggest to distinguish more explicitly in the discussion between association and agreement -particularly since cadence shows significant within-subject differences. Including in the analyses, and/or in the discussion, the percentage error (or relative difference) between the values obtained from the two tests, or the minimal detectable change between them, could more clearly show whether the observed differences are clinically meaningful.
Line 428: ‘particularly in clinical settings with time constraints’ is repeated twice.
Author Response
For research article
|
Response to Reviewer 1 Comments
|
||
|
1. Summary |
|
|
|
We sincerely thank Reviewer 1 for recognizing the relevance of our study, particularly regarding the potential of shorter walking tests to reduce patient burden and enhance clinical feasibility. Below, we provide a detailed, point-by-point response to each comment, including clarifications regarding the general evaluation. The specific lines where changes were made in the revised manuscript are indicated in brackets. All modifications in the revised version are highlighted in yellow for clarity.
|
||
|
2. 2. Questions for General Evaluation We have made efforts to improve the manuscript as suggested in this section. The details are provided below.
3. Point-by-point response to Comments and Suggestions for Authors |
||
|
Comment 1. Introduction The manuscript would benefit from a clearer articulation of how the present work advances beyond prior studies -beside the reduced duration of the test, for example, by emphasizing the longitudinal (1-2-year) aspect of the current work. I suggest the authors to briefly explain why they extracted and analyze much less parameters than those used in previous studies who investigated the 6MWT in PD with identical set-up, e.g.,: Bailo, G., Saibene, F. L., Bandini, V., Arcuri, P., Salvatore, A., Meloni, M., ... & Carpinella, I. (2024). Characterization of walking in mild Parkinson’s disease: Reliability, validity and discriminant ability of the six-minute walk test instrumented with a single inertial sensor. Sensors, 24(2), 662. Please briefly describe in the introduction the 6MWT/2MWT and 10 MWT -I mean briefly describe the procedure and the instructions to participants. Currently, it is not really clear for someone who is not a PD expert. Response 1: Point 1. We thank the reviewer for these insightful comments. We have now clarified in the Introduction that the present work extends previous studies by adopting a longitudinal design with repeated assessments at 1- and 2-year follow-ups (Lines 118–121). Although the main results focus on baseline data, the study was designed to capture within-subject changes over time and to better contextualize baseline findings, which have not been systematically explored in previous studies on PD. The follow-up data, reported in the Supplementary Materials, further supports the feasibility and longitudinal stability of the proposed approach. Regarding the second point, we focused on the core set of spatiotemporal parameters directly available in the software report. While the raw data includes step-by-step parameters, these are not presented in the report. We chose this approach because analyzing only the raw data would not have allowed us to confirm correspondence with the 6MWT report values or ensure comparability with the system’s internal computations. We have now clarified the definition of the “core set” of parameters in the Introduction (Lines 114-116) and the rationale for this choice in the Methods section (Lines 167-179) and Limits (Lines 502-509). Point 3. We have therefore expanded the Introduction to briefly describe the procedure and the instructions to participants for the 6MWT, 2MWT, and 10MeWT (Lines 73-79).
|
||
|
Comment 2. Methods Number of participants is not reported -it is only mentioned in the abstract and a different number is reported in the Supplementary materials as ‘original cohort’. Please specify what this means and report it in the main text of the manuscript. The sample size (n=43) is reasonable for exploratory work, yet a power analysis or sensitivity estimate would strengthen the justification. Please move Table 1 at the beginning of the Methods, when participants are described and add IPAQ scores to it -or explain why you did not report them in the Table nor analyzed them. Add to this Table also the subjectively reported falls. Line 211: unclear what are the “two methods” compared with the Bland-Altman method. From Figure 2 it seems this test was used to compare performance in the two tests’ length (6 vs 2 minutes long). Please clarify this in the methods. I suggest to add to the methods the description of the correlations with baseline characteristics of the patients Response 2: We thank the Reviewer for this helpful comment. Point 1. Number of participants. We have clarified the sample size in the Abstract. As explained in the Methods (paragraph: Study design and patient characteristics), the original cohort consisted of N = 62 participants. The analyses presented here were restricted to data from two of the three recruitment centers, for which complete data across all time points (baseline, 1-year, and 2-year follow-up) were available. No significant differences were found between the included study sample and the overall original cohort in terms of the demographic and clinical characteristics, see Table 1S. A statement regarding the adequacy of the sample size has been added to the Methods (Lines 133-135). Point 2. We have moved Table 1 immediately following the description of the participants and have also added IPAQ scores and the number of subjectively reported falls to Table 1 and Results (Lines 288-293). Point 3. We have clarified, in the Methods section and in the legend of Figure 2, that Bland–Altman and correlation analyses were performed to compare the 2′6MWT and the 6MWT. Comment 3. Results Why one participant was excluded based on distance covered? The authors say they excluded this patient because was not able to finish the task, but then only distance covered was reported. Please explain it further. The authors use non-parametric tests and correlations appropriately, but the lack of correction for multiple comparisons (beyond Bonferroni for Friedman tests) may risk type I error inflation. Why other collected baseline measures such as IPAQ scores were not correlated with the 2MWT and 6MWT? I suggest to include them as they may also impact the outcomes. How AR and TD subtypes were defined? Please specify No patient experienced freezing of gait during the gait trial? I suggest to specify it (and state whether your group did include freezers or not). How many participants performed the test again at 1 year and at 2 years post baseline assessment? Please add this information, and also please report in the Table about individual baseline values, also available data for these later assessments -whatever you collected. Please report confidence intervals for the mean difference resulting from the CCC analyses also in the main text and not just in Tables. Response 3: We thank the Reviewer for this helpful comment. We provide detailed, point-by-point responses to each issue below. Point 1. One participant was excluded because he was unable to complete the 2MWT/6MWT. The quality and temporal resolution of the sampling data are crucial for accurately calculating spatial gait parameters. We have clarified this in the Methods (Lines 262-263). Point 2. The description of the statistical procedures may have caused some misunderstanding. We have now better clarified in both the Methods section (Lines 246-249) and the table legends. Specifically, only for analyses involving three consecutive 2-minute segments, overall effects were assessed using Friedman tests, followed by Bonferroni-adjusted pairwise Wilcoxon signed-rank post-hoc comparisons for contrasts. Point 3. IPAQ scores Has been included in the correlation matrix. Point 4. The AR and TD subtypes were classified based on a clinical evaluation. Specifically, patients were categorized as TD if tremor was the predominant motor feature, and as AR if bradykinesia and rigidity were the predominant features. The AR and TD subtypes were classified based on the patients’ clinical records, reflecting a comprehensive clinical evaluation rather than relying solely on UPDRS-III motor scores obtained in the ON phase. This classification approach has now been clarified in the Methods section (Lines 191-195). Point 5. During the gait assessments, no participant exhibited FOG (evaluation in phase ON). FOG was mainly derived by UPDRS III and none of the patients experienced freezing episodes during the trial. This clarification has now been added to the Methods (Lines 197-199) and Results sections (Lines 288-93). Point 6. The number of participants with complete follow-up data is now reported in the corresponding tables in the Supplementary File, with n = 40 at the 1-year follow-up and n = 35 at the 2-year follow-up. A table presenting individual baseline values and the availability of follow-up data has been added to the Supplementary table 8S and discussed in the Results (Lines 397-399). Point 7. The 95%CI for the mean differences resulting from the CCC analyses is now included in the main text.
Comment 4. Discussion Line 328: The authors state they tested patients at early stage of disease but they had patients also with H&Y score of 3 considered moderate stage of disease. Please correct. I suggest to mention in the discussion that this shorter 2M test might be fine for early-stage PD or certain PD subtypes (e.g., non-freezers), but it might be insufficient to capture other aspects of the disease such as freezing of gait that becomes more common when the patient experience a little fatigue (i.e., with longer assessments). More advanced stages of PD may need another validation. The study relies on a single IMU (G-WALK). Although this is consistent with prior literature, the reliability of single-sensor placement should be discussed in relation to multi-sensor systems that may capture turning dynamics or asymmetry and provide richer pool of gait parameters. I suggest to add this to the discussion because it would strengthen the translational implications for clinical deployment. The findings show strong correlations between 2MWT and 6MWT parameters; however, correlation does not necessarily imply interchangeability. I suggest to distinguish more explicitly in the discussion between association and agreement -particularly since cadence shows significant within-subject differences. Including in the analyses, and/or in the discussion, the percentage error (or relative difference) between the values obtained from the two tests, or the minimal detectable change between them, could more clearly show whether the observed differences are clinically meaningful. Line 428: ‘particularly in clinical settings with time constraints’ is repeated twice. Response 4: We thank the reviewer for these valuable suggestions. Point 1. We have revised the text to clarify that patients were mainly tested at an early-to-mild stage of the disease (H&Y ≤ 2), with only one patient in the moderate stage (H&Y = 3). A sentence in Discussion has been corrected (Lines 405-406). Point 2. We agree that phenomena such as fatigue and freezing of gait can significantly influence performance in longer walking assessments and are more prevalent in patients at more advanced stages of PD. In our study, however, all participants were in the “ON” state and at an early-mild stage of the disease, with none exhibiting gait interruptions during the 6MWT. Therefore, validating the 2’6MWT by resampling data from the 6MWT is appropriate for this population and aligns with our study’s objective. Nonetheless, we recognize that further validation of the 2’6MWT is warranted in patients with more advanced PD, higher fatigue burden, or frequent freezing episodes. We have clarified this point in the Discussion (Lines 519-525). Point 3. We have included a discussion of the limitations (Lines 510-518) of using a single IMU. While consistent with previous literature, we acknowledge that multi-sensor systems can capture additional gait features such as dynamics and asymmetry, potentially providing a richer set of parameters. Point 4. This aspect was not clearly articulated in the Discussion; we have clarified this distinction (Lines 412-418). Our results show that for stride length and gait speed, the values obtained from the 2′6MWT and the 6MWT are strongly correlated and in agreement, indicating that these parameters can be considered effectively interchangeable between the two tests. In contrast, cadence exhibited small but systematic within-subject differences, warranting caution in the interpretation of data. We added the percentage errors (PE) to the text (Lines 351-352). Overall, the combination of linear regression (predictive validity), Concordance Correlation Coefficient (agreement), Bland-Altman analysis (bias and limits of agreement), and PE (relative measurement error) provides a comprehensive and methodologically sound assessment of the relationship between the 2'6MWT and 6MWT parameters. The repetition has been removed (point 5).
Response to Comments on the Quality of English Language |
||
|
Response: |
||
|
We have improved the overall flow of the English and corrected the discrepancies previously noted. Some sections have also been revised to read more smoothly.
|
||
|
5. Additional clarifications |
||
|
Response: |
||
|
None
|
||

Reviewer 2 Report
Comments and Suggestions for Authors
I read with great interest the study evaluating whether two minutes of walking can reliably measure gait parameters compared to the traditional six-minute walk. The study measured gait speed, stride length, and cadence using an IMU sensor, and revealed that a two-minute walk correlates strongly with six-minute walk. This paper addresses a clinically relevant and timely question, as shorter walking tests can reduce patient burden while improving clinical feasibility. However, I have several concerns and questions for clarification as outlined below.
Major Comments
- Methods (Lines 154–156): Please provide more detail on how gait speed was estimated from a single waist-mounted IMU sensor. Has this approach been validated in previous studies? The reported gait speeds range approximately 0.8–1.8 m/s. 1.8 m/s appear to be very fast, even for healthy controls. Additional information regarding the calculation and validation of gait speed would be helpful.
- Gait features: The study focused on gait speed and stride length outcomes. However, gait variability, arrhythmicity (commonly measured as stride-time variability), and asymmetry are key features of Parkinson’s disease gait and clinically important markers of motor deficits. The accuracy of these parameters was not addressed in the study. Including these PD-specific gait features, or acknowledging their omission as a limitation, would strengthen the clinical impact of the work.
- Results (Table 1): Please include UPDRS-III gait subscores and freezing-of-gait (FOG) scores additional demographic information for the cohort to help readers contextualize the findings.
- Discussion (Line 368): Gait speed measured from the two-minute walk was significantly higher than from the six-minute walk may not necessarily be due to fatigue. This difference could also stem from the manifestation of the “sequence effect” specific to PD (Iansek et al., 2006; Cui et al., 2025). Please discuss this alternative explanation.
Minor Comment
- Figure 1: The illustration of stride length is unclear. Stride length should be measured from heel contact to the next heel contact of the same foot; the figure does not appear to reflect this definition.
References
Iansek, R., Huxham, F., & McGinley, J. (2006). The sequence effect and gait festination in Parkinson disease: contributors to freezing of gait? Movement Disorders, 21(9), 1419–1424.
Cui, C., Je, G., Wilkins, K. B., Schulte, T., & Bronte-Stewart, H. M. (2025). Subthalamic Deep Brain Stimulation Alleviates the Gait Sequence Effect and Freezing of Gait in Parkinson’s Disease. Parkinsonism & Related Disorders, 108062.
Author Response
For research article
|
Response to Reviewer 2 Comments
|
||
|
1. Summary |
|
|
|
We sincerely thank Reviewer 1 for recognizing the clinical relevance and timeliness of our study, particularly regarding the potential of shorter walking tests to reduce patient burden and improve clinical feasibility. The reviewer’s suggestions have helped us enhance the quality and clarity of the manuscript. Below, we provide a detailed point-by-point response to each comment, including the questions regarding general evaluation. The lines where changes were made in the manuscript are indicated in brackets. All changes in the revised manuscript are highlighted in yellow for clarity.
|
||
|
2. 2. Questions for General Evaluation We have made efforts to improve the manuscript as suggested in this section. The details are provided below.
3. Point-by-point response to Comments and Suggestions for Authors |
||
|
Comment 1 (Methods, Lines 154–156): Please provide more detail on how gait speed was estimated from a single waist-mounted IMU sensor. Has this approach been validated in previous studies? The reported gait speeds range approximately 0.8–1.8 m/s. 1.8 m/s appear to be very fast, even for healthy controls. Additional information regarding the calculation and validation of gait speed would be helpful. Response 1: We thank the reviewer for this comment. We have expanded the Methods section (Lines 220-236) to provide additional details on how gait metrics were derived from the waist-mounted IMU (BTS G-Walk). Data were recorded and processed using the manufacturer’s G-Studio software according to the standard protocol. The exact algorithms of the G-Walk are proprietary and not publicly available. However, according to the manufacturer’s technical specifications, gait speed is calculated by integrating the accelerometer signal over each walking trial. These calculations are averaged over steady walking periods, automatically excluding the acceleration and deceleration phases at the beginning and end of the test. Gait speed measured with the G-Walk has been previously validated in patients with PD (Vítečková et al., 2020), showing good agreement with gold-standard gait analysis systems. This validation study reported a mean gait speed of 1.06 ± 0.30 m/s, the approximate 2.5th–97.5th percentiles correspond to 0.5–1.7 m/s (derived assuming a normal distribution, mean ± 2SD), which is closely matches with the range observed in our cohort. In our sample, the median gait speed was 1.3 m/s (IQR 1.2–1.5) at 2 minutes and 1.2 m/s (IQR 1.1–1.4) at 6 minutes, with individual values ranging overall from 0.7 to 1.8 m/s; only two participants reached 1.8 m/s. To better illustrate the variability in gait parameters and the handling of outliers, a section has now been included in the Results (Lines 308-312). We also clarified in the Methods (Lines 259-261) that all raw gait data were reviewed for potential outliers using boxplots and the Tukey method (values outside 1.5 × IQR from the first and third quartiles). Excluding these few extreme values (<5%) did not substantially alter the median values or the results of the statistical tests; therefore, all measurements were retained in the analysis.
Comment 2 (Gait features): The study focused on gait speed and stride length outcomes. However, gait variability, arrhythmicity, and asymmetry are key features of PD gait and clinically important markers of motor deficits. Including these PD-specific gait features, or acknowledging their omission as a limitation, would strengthen the clinical impact of the work. Response 2: We fully agree with this comment. This point was previously discussed only in the Supplementary Materials, but it is now highlighted in the manuscript. The spatiotemporal parameters reported in our study correspond to the outputs generated by the software. The manufacturer’s software provides summary metrics for each 6MWT trial, including mean gait speed, cadence, stride length, test duration, distance covered, pauses, and a General Symmetry Index (GSI). Other stride-by-stride metrics, such as gait variability or arrhythmicity, are not included in the standard outputs and would require a different test, which was beyond the scope of this study. We relied on these parameters, whose computational algorithms—including symmetry calculations—are proprietary and not accessible for custom processing. The GSI is computed over the entire walking trial to assess left–right symmetry based on stance and swing phases. Extracting shorter subsegments (e.g., the first 2 minutes) would have required reprocessing the raw data, which could potentially affect the accuracy of gait parameters. Consequently, symmetry metrics were reported only for the complete 6MWT. This methodological constraint is now explicitly described in the Methods (Lines 167-170) and Limitations (Lines 502-509).
Comment 3 (Results, Table 1): Please include UPDRS-III gait subscores and freezing-of-gait (FOG) scores additional demographic information for the cohort to help readers contextualize the findings. Response 3 We have updated Table 1 to include the UPDRS-III gait subscores (PIGD and FOG). The additional results from Table 1 are referenced in the Results section (Lines 288-293). No relevant comorbidities affecting gait were present in the included participants at baseline, as specified for exclusion criteria. Patients were evaluated in the ON phase, and no gait stops were recorded by the sensor. All relevant demographic information for the cohort is reported, and the age range has been added to provide further clarity on cohort characteristics. A supplementary table (Table 8S) including additional follow-up data has also been provided.
Comment 4 (Discussion, Line 368): Gait speed measured from the two-minute walk was significantly higher than from the six-minute walk may not necessarily be due to fatigue. This difference could also stem from the “sequence effect” specific to PD (Iansek et al., 2006; Cui et al., 2025). Please discuss this alternative explanation. Response 4: Thank you for this insightful suggestion. In our original manuscript, we focused on fatigue, as this was the aspect specifically addressed in outpatient settings, where time is often a limiting factor and patients may be particularly susceptible to fatigue or stress. However, we agree that the sequence effect should be mentioned as an additional explanation. We have now revised the Discussion (Lines 450-453) to include this alternative explanation for the observed decrease in gait speed during prolonged walking in PD participants, citing the suggested references.
Minor Comment Comment 5 (Figure 1): The illustration of stride length is unclear. Stride length should be measured from heel contact to the next heel contact of the same foot; the figure does not appear to reflect this definition. Response 5: We thank the reviewer for pointing this out. Figure 1 has been corrected to accurately depict stride length as the distance between successive heel strikes of the same foot.
|
||
|
4. Response to Comments on the Quality of English Language |
||
|
Response: |
||
|
We thank for the positive evaluation of the English quality.
|
||
|
5. Additional clarifications |
||
|
Response: |
||
|
None
|
||

Reviewer 3 Report
Comments and Suggestions for Authors
The paper presents an exploration of the possibility of using a 2-minute walk test compared to the traditional 6-minute test, with the aim of reducing testing time and facilitating analysis in subjects prone to fatigue. For this purpose, a sensorized version of the tests is used to compare the spatiotemporal parameters of the gait cycle between both protocols.
The manuscript is well written, and the methodology is clearly presented, making it reproducible. However, there are some details that should be further specified:
Between lines 98 and 100, a parenthesis is left open.
Between lines 101 and 103, it is not indicated for which type of patients the statement applies that inertial sensor–based systems provide a simple, fast, and clinically acceptable alternative for gait parameter measurement. This should be clarified according to the cited reference.
How were the initial gait segments handled? Were they excluded from the analysis or considered? This point is important, as gait acceleration and deceleration phases present different parameters from those measured during steady walking and could affect the results.
The quality of the graphics in Figure 2 should be improved. In panels A and B, it would also be useful to specify which line corresponds to the segmented trace and which to the non-segmented one.
It is very important to state whether the G-walk device has been validated for patients with Parkinson’s disease and provide evidence from a relevant study. Since the tests rely on gait parameters estimated with inertial sensors (mainly accelerometers), there may be issues when estimating these parameters in populations other than healthy subjects.
Author Response
For research article
|
Response to Reviewer 3 Comments
|
||
|
1. Summary |
|
|
|
We sincerely thank Reviewer 2 for his careful recognition of the clarity and reproducibility of the methodology, as well as the relevance of investigating a 2-minute walk test as a time-efficient alternative to the traditional 6-minute test for participants prone to fatigue. We have carefully considered the points raised regarding additional methodological details and provide detailed point-by-point responses below, including the questions regarding general evaluation. The lines where changes were made in the manuscript are indicated in brackets. All changes in the revised manuscript are highlighted in yellow for clarity.
|
||
|
2. 2. Questions for General Evaluation We have made efforts to improve the manuscript as suggested in this section. The details are provided below.
3. Point-by-point response to Comments and Suggestions for Authors |
||
|
Comment 1 Between lines 98 and 100, a parenthesis is left open. Response 1: We thank the reviewer for pointing this out. The parenthesis has been corrected.
Comment 2 Between lines 101 and 103, it is not indicated for which type of patients the statement applies that inertial sensor–based systems provide a simple, fast, and clinically acceptable alternative for gait parameter measurement. This should be clarified according to the cited reference. Response 2: We thank the reviewer for pointing this out. We have clarified in the revised manuscript that the statement refers specifically to patients with Parkinson’s disease (Introduction, Line 108).
Comment 3 How were the initial gait segments handled? Were they excluded from the analysis or considered? This point is important, as gait acceleration and deceleration phases present different parameters from those measured during steady walking and could affect the results. Response 3: We thank the reviewer for this important methodological point. We utilized the full 6MWT dataset provided by the G-Studio software. The software processes trunk acceleration and angular velocity signals using proprietary algorithms that automatically detect gait events via peak and zero-crossing analysis. Importantly, as detailed in the G-Walk manual (v.7.0.0), the software automatically omits the first and last walking paths from the data, corresponding to the initial acceleration and final deceleration phases, which are not representative of walking. This behavior is also evident in the raw data, which do not include these discarded segments. The system also identifies and removes steps associated with discontinuities during the walk. Consequently, only stable gait segments contribute to the calculation of spatiotemporal parameters, minimizing any bias. We have expanded the Methods section (Lines 220-236) to provide additional details on how gait metrics were derived from the waist-mounted IMU (BTS G-Walk).
Comment 4 The quality of the graphics in Figure 2 should be improved. In panels A and B, it would also be useful to specify which line corresponds to the segmented trace and which to the non-segmented one. Response 4: The quality of Figure 2 has been improved, and all elements are now clearly labeled. The figure legend has also been enhanced for clarity. Plot colors have been adjusted for better distinction: white for stride length and black for gait speed. The label of Panels A and C now indicate that the regression analyses refer to 2’6MWT vs. total 6MWT, with a reference line (y = x) for comparison of stride length and gait speed data. The label of Panels B and D now indicate that these panels show Bland-Altman plots between 2 and 6 minutes, depicting the mean and differences for stride length and gait speed data.
Comment 5 It is very important to state whether the G-walk device has been validated for patients with Parkinson’s disease and provide evidence from a relevant study. Since the tests rely on gait parameters estimated with inertial sensors (mainly accelerometers), there may be issues when estimating these parameters in populations other than healthy subjects. Response 5: We thank the reviewer for this important comment. The G-Walk system has been previously compared with gold-standard gait analysis methods in both healthy controls and patients with PD. In particular, the study by Vítečková and colleagues demonstrated good agreement for cadence, stride duration, gait speed, and stride length, indicating that the G-Walk can reliably capture the temporal and spatial gait parameters relevant to the 6MWT. Systematic differences have been reported for gait phase durations (stance, swing, single and double support) in PD patients, likely due to the trunk-based gait event detection algorithm. Although these differences are less critical in the context of the 6MWT—our study focused on averaged gait metrics over the entire walking trial rather than detailed phase-by-phase analysis—the known limitations of trunk-mounted inertial sensors have been discussed in the revised manuscript. This clarification, together with supporting validation references (n=2), has been added to the Methods (Lines 234-236).and Limitations sections (Lines 510-518).
4. Response to Comments on the Quality of English Language |
||
|
Response: |
||
|
We thank for the positive evaluation of the English quality.
|
||
|
5. Additional clarifications |
||
|
None |
||

Round 2
Reviewer 1 Report
Comments and Suggestions for Authors
The authors have made some revisions, but a number of concerns remain only partially addressed. I believe further clarification and refinement are needed before the manuscript is ready for publication. My detailed comments follow below.
Note that the figures were not included in the manuscript, so I could not review them.
Introduction
The introduction still doesn’t flow and it feels confused and partial. The aims of the study are not clearly stated. In addition, no previous study using wearable sensors and no previous study using the same set-up are discussed. It is not stated in the whole introduction that this is the first time the 2’6MWT is assessed and compared to the 6MWT in PD -was it compared in other populations?
The authors added only partially what I asked in my previous round of revision. Even when they added information, this was not harmonized with the sentences before/after, so the introduction feels not really coherent -there are repetitions, logic gaps. Please harmonize the whole section. Below there are specific examples:
Line 75: I suggest to delete: ‘The test requires approximately 10 minutes’ -it is confusing to the readers given it is called the 6 min test. Or explain why it takes longer to administer.
Line 80: with the addition of the explanations of the other tests, the sentence becomes unclear: “However, its applicability in this population is still limited by the time of examiners and the patients for completion.” Please clarify that the sentence refers to the 6MWT.
Line 88: Please adjust this sentence: ‘In contrast, the psychometric properties of the 2MWT and 10MeWT [13] tests are still unknown for this population’. The paragraph before doesn’t say that all the psychometric properties of the 6MWT are known..
Line 96: the sentence: ‘in certain neurodegenerative and neuromuscular diseases’ please specify that this is not PD otherwise the sentence after -starting at line 102 appears entirely in contradiction.
Lines 100-102: Relocate this sentence elsewhere -logically it doesn’t belong here and also it is unclear what does it mean to conduct the two tests simultaneously? ‘Given their similar protocols and settings, conducting the 2MWT and 6MWT simultaneously can reduce participant fatigue and improve compliance’. EDIT: I understand from the revised methods that the authors likely meant that the 2MWT was derived from the full 6MWT. This is crucial aspect of the design and should be further discussed here: are there other studies using the same approach? Is it a procedure never used before? Is it standard procedure to acquire both tests separately? The issue should be introduced here and then further addressed in the discussion, e.g., as a potential limitations of the current results.
Lines 102-108: ‘However, in people with PD, gait speed tends to decline over the course of a 6MWT, with participants maintaining only ~76% of their maximal gait speed [21], largely due to hypokinesia, raising concerns about information redundancy, especially as disease severity increases [22]. In addition, the 6MWT performed with wearable inertial devices containing accelerometer and gyroscopic sensors, offers a fast, simple, clinically feasible method for real-time, objective assessment of gait quality in patients with PD’. In the first sentence the authors state the information is redundant, and in the second sentence they argue that the 6MWT is a fast test. Please clarify this apparent incongruency. Also, if the data shows that across the 6 minutes, gait speed tends to decrease, this suggests the 6M test is needed and may provide valuable information. Please clarify this issue further.
Lines 118-121: ‘The study therefore extends previous research by using a longitudinal design with assessments at baseline, 1 year, and 2 years to explore within-subject changes in gait performance over time’. This sentence was added without any prior explanation or mentioning of the importance of longitudinal studies -results from longitudinal studies on the 6MWT are not presented -if any. I suggest to add why longitudinal monitor is important and stress innovation (if present) with previous studies. Stating that ‘the study extends previous research’ when the previous research is not clearly summarized does not really clarify the motivation of the current work. Please extensively revise the introduction accordingly.
Methods
Study design and patient characteristics
The reporting of the sensitivity analysis is unclear. As written, it sounds as though an effect size of d ≈ 0.44 was observed, whereas this likely refers to the minimum detectable effect size given n = 43, α = 0.05, and power = 0.80. Please clarify that this analysis indicates the study was powered to detect within-subject effects of at least moderate magnitude (e.g., d ≥ 0.44), and specify which statistical test or model the sensitivity analysis was based on.
The information that all participants were tested in the ON condition is not included. Please specify it.
In the report answering to my previous queries the authors wrote: Point 2. We have moved Table 1 immediately following the description of the participants. This is not true as the table is still at the beginning of the results section. Please comply with this requirement or at least add a reference to this table in this section of the manuscript.
Please report also in this section (and not only in supplementary tables) the n of the participants who came back at the 1 year and 2 years follow-up.
Clinical and sensor-based assessment
Line 157: ‘During the annual visits..’. I suggest to change into: Assessments were conducted during annual visits for three consecutive years (i.e., baseline, 1 year and 2-years Follow-up)..
Line 161-162: The use of the term “ON phase” is unclear for participants not treated with L-DOPA. By definition, the ON phase refers to periods when dopaminergic medication is active. If some patients were unmedicated, please clarify how their assessment phase was defined, e.g., were they evaluated in their best functional state, or were they taking other dopaminergic agents? The terminology should be adjusted accordingly to avoid confusion.
Line 198-199: ‘FOG was not reported as occurring in daily life but was observed during the clinical assessment’. Please clarify. The authors wrote in their comments that none of the participants showed FOG during the assessments. Also clarify what this sentence means -as an alternative to the questionnaire or in addition to it?
Statistical analysis
Line 246: ‘For comparisons across the three consecutive 2-minute segments (segments 1–3)’ please add the clarification: For comparisons across the three consecutive 2-minute segments (segments 1–3) of the 6MWT..
Results
Participants characteristics
Line 289: ‘Freezing gait was largely absent during trial’: please clarify the sentence. No participant experienced FOG, no? In the table the score is 0. If so, then I suggest to change the sentence accordingly.
Comparison of 2'6MWT and 6MWT Parameters
Line 307: No gait stops were recorded. Please clarify, e.g., during the gait trials no gait stops were recorded (i.e., no FOG episodes were detected, and no voluntary stops).
Comparison of 2'6MWT and 6MWT Parameters
The Passing-Bablok regression is mentioned as yielding similar results, but the corresponding data are not shown. Given that this method is less commonly applied in gait analysis and assesses agreement differently from standard regression, it would be useful to provide at least summary parameters (e.g., slope, intercept, confidence intervals) or include these results in the Supplementary Material for transparency.
Line351: what is PE? Please explain all acronyms.
Table 4: what is the meaning of the asterisks (* vs **)? Please add this to the figure legend.
Also this sentence is unclear: ‘the associations with motor scores and cognitive status were considered, as showed the most consistent correlations. Specifically, greater motor impairment (higher PIGD) and lower cognitive function (lower MMSE, while MoCA in this cohort did not show significant correlations) were associated with reduced walking performance (distance covered, gait speed, and step length) across both time points’. From the table it seems many correlations were significant. Please clarify how ‘most consistent correlations’ were assessed. Also it is unclear which ‘both time points’ the authors are referring to: isn’t this the baseline performance? Or you mean the 2’6 vs 6MWT? Please clarify.
Discussion
Lines 424-428: I suggest to move this paragraph to the introduction to introduce previous similar and related works and more clearly introduce what the innovation of the current work is.
Lines 448-449: This sentence ‘Gait speed was greater during the initial 2 minutes of the 6MWT, without influencing its correlation with the total 6MWT performance’ seems incompatible to the sentence in lines 459-461: ‘Analyzing the 6MWT in three consecutive 2-minute segments revealed a slight decrease in distance and cadence during the first segment, while stride length and gait speed remained stable’. Please explain what may have originated this difference -e.g., difference in pacing?
Line 476: ‘Moreover, in the present cohort, all but one participant successfully completed the first 2 minutes of the 6MW’. Please clarify as you wrote above that the excluded participant stopped at 2.39 minutes -so it seems he did not complete the 6MWT, but he did complete the 2’6MWT.
Lines 493-494: ‘Overall, these results support the use of the 2’6MWT as a representative measure of the full test, eliminating the need for a separate 2MWT’. I think this should be clarified. 1. It might be that parameters might change if the participant knows in advance that the test is overall shorter (so the 2MWT might still be useful, or the proposed conclusion must be directly tested by directly comparing results from these two tests -EDIT: I see that you point this out in the limitations so perhaps consider to delete this sentence). 2. If the authors think this is something that can be useful, I mean to only evaluate the first two minutes of a 6 minutes test (so still recording the full 6 minutes), please explain why, and provide examples of situations that might benefit from this approach.
Comments on the Quality of English LanguageThe language was appropriate. Some sentences were confusing but more form a conceptual than grammatical perspective.
Author Response
For research article
|
Response to Reviewer 1 Comments
|
||
|
1. Summary |
|
|
|
Thank you very much for your comments and suggestions, we greatly appreciate the time you dedicated to reviewing the revised version of our work. We have carefully considered your observations and agree that many of your points can significantly improve the manuscript. We recognize that our previous revisions may have inadvertently disrupted the logical flow of the manuscript, and we are grateful for the detailed guidance provided to improve its coherence and clarity. The lines where changes were made in the manuscript are indicated in brackets. All changes in the revised manuscript are highlighted in yellow for clarity.
2. Questions for General Evaluation We had revised the manuscript, in particular, we had addressed the introduction by incorporating the additional context you suggested to provide a more complete background. We also note your comment regarding the figures, now included in the revised version for full review. |
||
|
|
||
|
3. Point-by-point response to Comments and Suggestions for Authors |
||
|
Comments 1: Introduction The introduction still doesn’t flow and it feels confused and partial. The aims of the study are not clearly stated. In addition, no previous study using wearable sensors and no previous study using the same set-up are discussed. It is not stated in the whole introduction that this is the first time the 2’6MWT is assessed and compared to the 6MWT in PD -was it compared in other populations? The authors added only partially what I asked in my previous round of revision. Even when they added information, this was not harmonized with the sentences before/after, so the introduction feels not really coherent -there are repetitions, logic gaps. Please harmonize the whole section. Below there are specific examples: Line 75: I suggest to delete: ‘The test requires approximately 10 minutes’ -it is confusing to the readers given it is called the 6 min test. Or explain why it takes longer to administer. Line 80: with the addition of the explanations of the other tests, the sentence becomes unclear: “However, its applicability in this population is still limited by the time of examiners and the patients for completion.” Please clarify that the sentence refers to the 6MWT. Line 88: Please adjust this sentence: ‘In contrast, the psychometric properties of the 2MWT and 10MeWT [13] tests are still unknown for this population’. The paragraph before doesn’t say that all the psychometric properties of the 6MWT are known.. Line 96: the sentence: ‘in certain neurodegenerative and neuromuscular diseases’ please specify that this is not PD otherwise the sentence after -starting at line 102 appears entirely in contradiction. Lines 100-102: Relocate this sentence elsewhere -logically it doesn’t belong here and also it is unclear what does it mean to conduct the two tests simultaneously? ‘Given their similar protocols and settings, conducting the 2MWT and 6MWT simultaneously can reduce participant fatigue and improve compliance’. EDIT: I understand from the revised methods that the authors likely meant that the 2MWT was derived from the full 6MWT. This is crucial aspect of the design and should be further discussed here: are there other studies using the same approach? Is it a procedure never used before? Is it standard procedure to acquire both tests separately? The issue should be introduced here and then further addressed in the discussion, e.g., as a potential limitations of the current results. Lines 102-108: ‘However, in people with PD, gait speed tends to decline over the course of a 6MWT, with participants maintaining only ~76% of their maximal gait speed [21], largely due to hypokinesia, raising concerns about information redundancy, especially as disease severity increases [22]. In addition, the 6MWT performed with wearable inertial devices containing accelerometer and gyroscopic sensors, offers a fast, simple, clinically feasible method for real-time, objective assessment of gait quality in patients with PD’. In the first sentence the authors state the information is redundant, and in the second sentence they argue that the 6MWT is a fast test. Please clarify this apparent incongruency. Also, if the data shows that across the 6 minutes, gait speed tends to decrease, this suggests the 6M test is needed and may provide valuable information. Please clarify this issue further. Lines 118-121: ‘The study therefore extends previous research by using a longitudinal design with assessments at baseline, 1 year, and 2 years to explore within-subject changes in gait performance over time’. This sentence was added without any prior explanation or mentioning of the importance of longitudinal studies -results from longitudinal studies on the 6MWT are not presented -if any. I suggest to add why longitudinal monitor is important and stress innovation (if present) with previous studies. Stating that ‘the study extends previous research’ when the previous research is not clearly summarized does not really clarify the motivation of the current work. Please extensively revise the introduction accordingly. |
||
|
Response 1: We sincerely thank the reviewer for this comprehensive and insightful evaluation of the introduction, mentioned above, we recognize that our previous revisions may have inadvertently disrupted the logical flow of the manuscript. In response, we have revised the text, specifically: - We acknowledge that we had not sufficiently emphasized the novel methodological aspect of our study, specifically, that the 2'6MWT is derived from a single 6MWT session using wearable sensors, rather than being conducted as a separate, standalone test. This distinction is fundamental to understanding both the innovation and the limitations of our approach (Lines 105-125). - The sentence on wearable sensors has been edited in light of the references we considered valid for this purpose. - Finally, the rationale for a longitudinal design is strengthened, clarifying its importance for monitoring within-subject changes in gait performance over time (Lines 119-121).
Additional suggestions related to the remaining comments were considered and/or integrated.
Comments 2: Methods Study design and patient characteristics The reporting of the sensitivity analysis is unclear. As written, it sounds as though an effect size of d ≈ 0.44 was observed, whereas this likely refers to the minimum detectable effect size given n = 43, α = 0.05, and power = 0.80. Please clarify that this analysis indicates the study was powered to detect within-subject effects of at least moderate magnitude (e.g., d ≥ 0.44), and specify which statistical test or model the sensitivity analysis was based on. The information that all participants were tested in the ON condition is not included. Please specify it. In the report answering to my previous queries the authors wrote: Point 2. We have moved Table 1 immediately following the description of the participants. This is not true as the table is still at the beginning of the results section. Please comply with this requirement or at least add a reference to this table in this section of the manuscript. Please report also in this section (and not only in supplementary tables) the n of the participants who came back at the 1 year and 2 years follow-up. Clinical and sensor-based assessment Line 157: ‘During the annual visits..’. I suggest to change into: Assessments were conducted during annual visits for three consecutive years (i.e., baseline, 1 year and 2-years Follow-up).. Line 161-162: The use of the term “ON phase” is unclear for participants not treated with L-DOPA. By definition, the ON phase refers to periods when dopaminergic medication is active. If some patients were unmedicated, please clarify how their assessment phase was defined, e.g., were they evaluated in their best functional state, or were they taking other dopaminergic agents? The terminology should be adjusted accordingly to avoid confusion. Line 198-199: ‘FOG was not reported as occurring in daily life but was observed during the clinical assessment’. Please clarify. The authors wrote in their comments that none of the participants showed FOG during the assessments. Also clarify what this sentence means -as an alternative to the questionnaire or in addition to it? Statistical analysis Line 246: ‘For comparisons across the three consecutive 2-minute segments (segments 1–3)’ please add the clarification: For comparisons across the three consecutive 2-minute segments (segments 1–3) of the 6MWT. Response 2. Methods Point 1. Study design and patient characteristics - We appreciate the reviewer’s observation and agree that the previous wording on sensitivity analysis may have been misleading. The text has been rewritten (Lines 134-137). - We believe that presenting Table 1 at the beginning of the Results section aligns with standard reporting practices. Nevertheless, we have now added explicit references to Table 1 in the Methods. - The n of the participants who came back at the 1 year and 2 years follow-up has been reported (Lines 138-139). Point 2. Clinical and sensor-based assessment - All participants received dopaminergic treatment (including L-DOPA and dopamine agonists) and were evaluated during their ON phase (Lines 169-170). - We clarified that FOG was not assessed through self-reported questionnaires, as was the case for fall history. Rather, FOG was evaluated exclusively during the clinical examination using the UPDRS Part III “Freezing of Gait” item, and no participant showed FOG during testing (Lines 207-210).
Additional suggestions related to the remaining comments were carefully considered and have been incorporated.
Comments 3: Results Participants characteristics Line 289: ‘Freezing gait was largely absent during trial’: please clarify the sentence. No participant experienced FOG, no? In the table the score is 0. If so, then I suggest to change the sentence accordingly. Comparison of 2'6MWT and 6MWT Parameters Line 307: No gait stops were recorded. Please clarify, e.g., during the gait trials no gait stops were recorded (i.e., no FOG episodes were detected, and no voluntary stops). Comparison of 2'6MWT and 6MWT Parameters The Passing-Bablok regression is mentioned as yielding similar results, but the corresponding data are not shown. Given that this method is less commonly applied in gait analysis and assesses agreement differently from standard regression, it would be useful to provide at least summary parameters (e.g., slope, intercept, confidence intervals) or include these results in the Supplementary Material for transparency. Line351: what is PE? Please explain all acronyms. Table 4: what is the meaning of the asterisks (* vs **)? Please add this to the figure legend. Also this sentence is unclear: ‘the associations with motor scores and cognitive status were considered, as showed the most consistent correlations. Specifically, greater motor impairment (higher PIGD) and lower cognitive function (lower MMSE, while MoCA in this cohort did not show significant correlations) were associated with reduced walking performance (distance covered, gait speed, and step length) across both time points’. From the table it seems many correlations were significant. Please clarify how ‘most consistent correlations’ were assessed. Also it is unclear which ‘both time points’ the authors are referring to: isn’t this the baseline performance? Or you mean the 2’6 vs 6MWT? Please clarify. Response 3. Results Point 1. Participant characteristics To further clarify, gait stops (interruptions of walking) were detected using the sensor-based 6MWT recording system (Methods, Lines 178 and Results: 317-318), whereas fall history was assessed via a self-reported question referring to the month preceding the motor assessment (Methods, Lines 208-210 and Results, Line 300). The assessment of these parameters has now been clearly described in the revised manuscript. Point 2. Comparison of 2'6MWT and 6MWT Parameters - Passing-Bablok summary parameters have now been added to the Supplementary Material (table 3S). - The acronym “PE” has been fully spelled out. - The asterisk notation in the table was already explained in the legend of Table 4. Nevertheless, in light of the reviewer’s comment, the entire paragraph discussing the table’s results has been revised (Lines 377-382). Specifically, since this is an exploratory analysis, we focused our attention on variables showing at least two significant correlations between the 2’6MWT and 6MWT, as already specified in the Methods section.
Comments 4: Discussion Lines 424-428: I suggest to move this paragraph to the introduction to introduce previous similar and related works and more clearly introduce what the innovation of the current work is. Lines 448-449: This sentence ‘Gait speed was greater during the initial 2 minutes of the 6MWT, without influencing its correlation with the total 6MWT performance’ seems incompatible to the sentence in lines 459-461: ‘Analyzing the 6MWT in three consecutive 2-minute segments revealed a slight decrease in distance and cadence during the first segment, while stride length and gait speed remained stable’. Please explain what may have originated this difference -e.g., difference in pacing? Line 476: ‘Moreover, in the present cohort, all but one participant successfully completed the first 2 minutes of the 6MW’. Please clarify as you wrote above that the excluded participant stopped at 2.39 minutes -so it seems he did not complete the 6MWT, but he did complete the 2’6MWT. Lines 493-494: ‘Overall, these results support the use of the 2’6MWT as a representative measure of the full test, eliminating the need for a separate 2MWT’. I think this should be clarified. 1. It might be that parameters might change if the participant knows in advance that the test is overall shorter (so the 2MWT might still be useful, or the proposed conclusion must be directly tested by directly comparing results from these two tests -EDIT: I see that you point this out in the limitations so perhaps consider to delete this sentence). 2. If the authors think this is something that can be useful, I mean to only evaluate the first two minutes of a 6 minutes test (so still recording the full 6 minutes), please explain why, and provide examples of situations that might benefit from this approach. Response 4. Discussion Point 1. We agree with the reviewer that it is appropriate to start by highlighting the novelty of the study (Lines 415-430). However, we have chosen to retain part of the concept expressed in previous lines 424–428, as it emphasizes the importance of sensor-based measurements in obtaining objective data without the need for a retest, thus justifying the use of a single section for the analyses (Lines 433-438). Point 2. We thank the reviewer for the opportunity to discuss this finding at a speculative level (Lines 476-481). While it may appear contradictory that participants were faster during the initial 2 minutes of the 6MWT but no significant differences were observed when analyzing consecutive 2-minute segments, this can be explained: Statistically, comparing the first 2 minutes with the total 6MWT involves only two aggregated values, which allows small differences to be detected without correction for multiple comparisons. In contrast, when the 6MWT is segmented into multiple 2-minute intervals, a repeated-measures test (Friedman test) is applied, and post-hoc comparisons require adjustment for multiple testing. As a result, small differences between consecutive segments are “spread out” and not consistent enough to reach significance after correction. Clinically, participants may start at a higher pace due to motivation or a “pacing” effect. After the initial minutes, they stabilize at a more consistent speed to sustain effort throughout the 6MWT. Thus, the initial trend is detectable in the aggregate 2-minute vs total comparison but not in the segmental analysis. Point 3. For analytical consistency, we included only participants who completed the entire 6MWT, as the 2MWT values were extracted directly from the continuous test. All participants therefore performed the 2’6MWT; however, one participant was excluded from the analysis because they stopped after the first 2 minutes. These data further support the feasibility and clinical relevance of shorter walking assessments even in individuals with greater motor limitations. The sentence has been rewritten accordingly (Lines 492-496). Point 4. We sincerely thank the reviewer for this comment and hope we have correctly understood it. Our results are not intended to suggest replacing the 2MWT with the 2’6MWT. Rather, they indicate that the first 2 minutes provide a reliable reflection of 6MWT, supporting their potential use when time constraints or patient limitations prevent completion of the test. Accordingly, we have rephrased the sentence to avoid overstatement and to emphasize the exploratory nature of this interpretation. We have also included examples of potential applications of this approach (Lines 512-517). We believe that the limitations should still be maintained for clarity, acknowledging that performing a separate 2MWT could theoretically yield differences, even if the 2’6MWT appears representative in our cohort.
|
||
|
4. Response to Comments on the Quality of English Language |
||
|
Point 1: |
||
|
Response 1: We thank the reviewer for the feedback regarding the quality of English.
|
||
|
5. Additional clarifications |
||
|
Lexical adjustments and concise rewording have been made directly in the text.
|
||

Reviewer 2 Report
Comments and Suggestions for Authors
The authors have thoroughly addressed the questions and concerns in this revision. I have no further questions.
Author Response
Comments and Suggestions for Authors
The authors have thoroughly addressed the questions and concerns in this revision. I have no further questions.
Response: We thank the reviewer for their careful evaluation and positive feedback. We appreciate the acknowledgment that our revisions have addressed the questions and concerns raised.